# *Drosophila* pVALIUM10 TRiP RNAi lines cause undesired silencing of Gateway-based transgenes

Dimitrije Stanković[1], Gábor Csordás[2] , Mirka Uhlirova[1]

**Post-transcriptional gene silencing using double-stranded RNA has revolutionized the field of functional genetics, allowing fast and easy disruption of gene function in various organisms. In *Drosophila*, many transgenic RNAi lines have been generated in large-scale efforts, including the *Drosophila* Transgenic RNAi Project (TRiP), to facilitate in vivo knockdown of virtually any *Drosophila* gene with spatial and temporal resolution. The available transgenic RNAi lines represent a fundamental resource for the fly community, providing an unprecedented opportunity to address a vast range of biological questions relevant to basic and biomedical research fields. However, caution should be applied regarding the efficiency and specificity of the RNAi approach. Here, we demonstrate that pVALIUM10-based RNAi lines, representing ~13% of the total TRiP collection (1,808 of 13,410 pVALIUM TRiP–based RNAi lines), cause unintended off-target silencing of transgenes expressed from Gateway destination vectors. The silencing is mediated by targeting attB1 and attB2 sequences generated via site-specific recombination and included in the transcribed mRNA. Deleting these attB sites from the Gateway expression vector prevents silencing and restores expected transgene expression.**

## Introduction

dsRNA-mediated post-transcriptional gene silencing, also known as RNAi, represents an ancient antiviral defense mechanism that has been harnessed as a powerful tool for reverse functional genetics in various cultured cells, and plant and animal model systems (Kennerdell & Carthew, 1998; Lohmann et al, 1999; Wargelius et al, 1999; Wianny & Zernicka-Goetz, 2000; Zamore et al, 2000). The dsRNA silencing relies on a highly conserved RNase III enzyme called Dicer that recognizes and processes dsRNA into small 21- to 23-bp segments (siRNA). Upon loading into a multi-protein RNA-induced silencing complex, the guide strand facilitates pairing with complementary on-target RNAs, which are subsequently degraded by the action of Argonaut proteins (Tomari & Zamore, 2005). In *Drosophila*, spatiotemporally controlled RNAi knockdown can be achieved by expressing dsRNA using transgenic binary expression systems, such as Gal4/UAS (Brand & Perrimon, 1993). The initial transgenic RNAi lines were generated using plasmids with ligated inverted repeats transcribed from one (Kennerdell & Carthew, 1998) or two (Giordano et al, 2002) opposite promoters. However, these vectors were unstable and not well tolerated by the bacterial strains commonly used for cloning (Hagan & Warren, 1983; Piccin et al, 2001), a problem later solved by the introduction of an intron spacer between the repeats (Lee & Carthew, 2003; Bao & Cagan, 2006). This step-by-step progress led to the development of the pVALIUM (vermilion-attB-loxP-intron-UAS-MCS), a universal vector system for transgenic RNAi library generation. The pVALIUM RNAi vectors contain an eye-color selection marker (*vermillion*), an attB site enabling phiC31-mediated site-directed integration into predefined genomic locations, *white* or *ftz* introns promoting the processing of dsRNA, and a cassette of two UAS pentamers, one of which is flanked by loxP sites to allow tuning of the dsRNA expression levels, and multiple cloning sites (MCSs) (Ni et al, 2008).

The highly efficient pVALIUM10-based transgenic RNAi stocks produced within the *Drosophila* Transgenic RNAi Project (TRiP; https://fgr.hms.harvard.edu/fly-in-vivo-rnai) at Harvard Medical School (HMS) https://fgr.hms.harvard.edu represent an invaluable resource for the *Drosophila* community (Perkins et al, 2015). At the moment of writing, 1,808 transgenic TRiP RNAi stocks based on the pVALIUM10 are available at the Bloomington *Drosophila* Stock Center (BDSC; https://bdsc.indiana.edu/index.html) (Perrimon et al, 2010). The pVALIUM10 vector carries *gypsy* insulators that markedly enhance knockdown efficiency, along with two inverted "attR1-selection-attR2" cassettes that facilitate cloning of dsRNA fragments via site-specific Gateway in vitro recombineering (Ni et al, 2009; Reece-Hoyes & Walhout, 2018). A desired single fragment, 400–600 bp in length and flanked by attL1 and attL2 recombination sites, can be easily moved from an entry vector into a pVALIUM10 destination vector via recombination between attL and attR sites catalyzed by LR Clonase, producing a vector for the UAS-driven expression of dsRNA (Ni et al, 2009).

[1]Institute for Genetics and Cologne Excellence Cluster on Cellular Stress Responses in Aging-Associated Diseases (CECAD), University of Cologne, Cologne, Germany [2]Institute of Genetics, Biological Research Centre of the Eötvös Loránd Research Network, Szeged, Hungary

Correspondence: mirka.uhlirova@uni-koeln.de

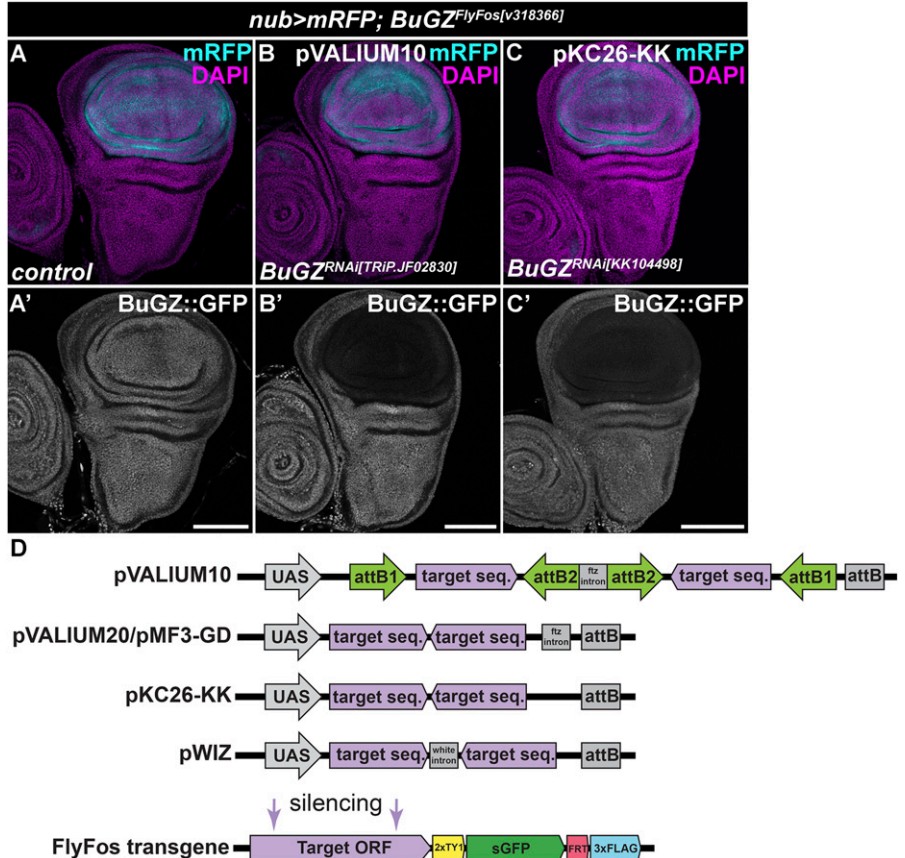

**Figure 1. pVALIUM10 and KK RNAi lines cause efficient silencing.**

**(A, B, C)** Expression of pVALIUM10 *BuGZ^RNAi[TRIP.JF02830]* (B, B') and KK-based *BuGZ^RNAi[KK104498]* transgenic RNAi lines (C, C') using the *nubbin-Gal4, UAS-myr-mRFP* driver (*nub > mRFP*, light blue) causes a marked reduction in a BuGZ^FlyFos[v318366] transgenic protein in the pouch region of wing discs (WDs) relative to the hinge and notum, and control WD (A, A'). Micrographs show projections of multiple confocal sections of WDs dissected from third instar larvae 7 d AEL. The multi-epitope–tagged BuGZ^FlyFos [v318366] protein was visualized by immunostaining with an anti-GFP antibody (white). Nuclei were counterstained with DAPI (magenta). Scale bars: 100 *μm*. **(D)** Schematic representation of various vectors used for cloning and overexpression of dsRNA. In the absence of the target-specific antibody, the RNAi efficacy can be assessed with the help of FlyFos transgenes expressing multi-epitope–tagged proteins under the control of the endogenous regulatory sequences.

The Gateway cloning technology is also routinely used to generate transgenic constructs for the expression of epitope-tagged or untagged proteins, taking advantage of the *Drosophila* Gateway plasmid collection at the *Drosophila* Genomics Resource Center (DGRC; https://dgrc.bio.indiana.edu/). Importantly, as with pVALIUM10 dsRNA plasmids, production of Gateway expression vectors relies on the recombination between attL and attR sites resulting in the formation of attB1 and attB2 sites.

Here, we demonstrate that the pVALIUM10-derived RNAi lines cause undesirable knockdown of transgenic reporters and overexpression lines generated with the help of the Gateway cloning technology. They do so by targeting the attB sites that are part of the transcribed mRNA. Deletion of the attB1 and attB2 sequences from the Gateway expression destination vector abrogates this effect and restores expected transgene expression levels.

# Results

## Unexpected, off-target silencing of the RNase H1::GFP reporter by multiple pVALIUM10 RNAi lines

In vivo RNAi-based genetic screens have proved very successful in identifying novel genes involved in developmental and pathologic processes in a variety of *Drosophila* somatic tissues when expressed using tissue-specific Gal4 driver lines (Lesch et al, 2010; Neely et al, 2010; Saj et al, 2010; Port et al, 2011; Zeng et al, 2015; Pletcher et al, 2019; Rotelli et al, 2019; Zhou et al, 2019; Graca et al, 2021; Rylee et al, 2022). Hence, we embarked on a candidate genetic screen aimed at identifying factors controlling genome stability. During the pre-screening of potential candidates, we observed that the expression of pVALIUM10-based *BuGZ^RNAi[TRiP.JF02830]* RNAi line in the pouch region of the developing third instar wing discs, using the *nubbin-Gal4, UAS-mRFP* (*nub > mRFP*) driver line, efficiently suppressed the multi-epitope–tagged BuGZ transgene (*BuGZ^FlyFos[v318366]*) (Sarov et al, 2006) (Fig 1A, B, and D). A similar reduction was observed using an independent *BuGZ^RNAi[KK104498]* line from the Vienna *Drosophila* Resource Center (VDRC) KK RNAi library (Fig 1C and D). Encouraged by these results, both BuGZ RNAi lines were included in the candidate genetic screen that was based on the expression of selected dsRNAs with the *nub > mRFP* driver. As readout for genome stability, we used the ubiquitously driven RNase H1::GFP that was generated by recombining the *RNase H1* coding sequence into the pUWG vector from the *Drosophila* Gateway plasmid collection. This resulted in the expression of a C-terminally GFP-tagged RNase H1 under the control of the poly-ubiquitin promoter (Fig 2A and B). The RNase H1 enzyme recognizes and resolves R-loops, DNA:RNA hybrids, that emerge during transcription and represent a threat to genome stability when unresolved (Santos-Pereira & Aguilera, 2015; Crossley et al, 2019).

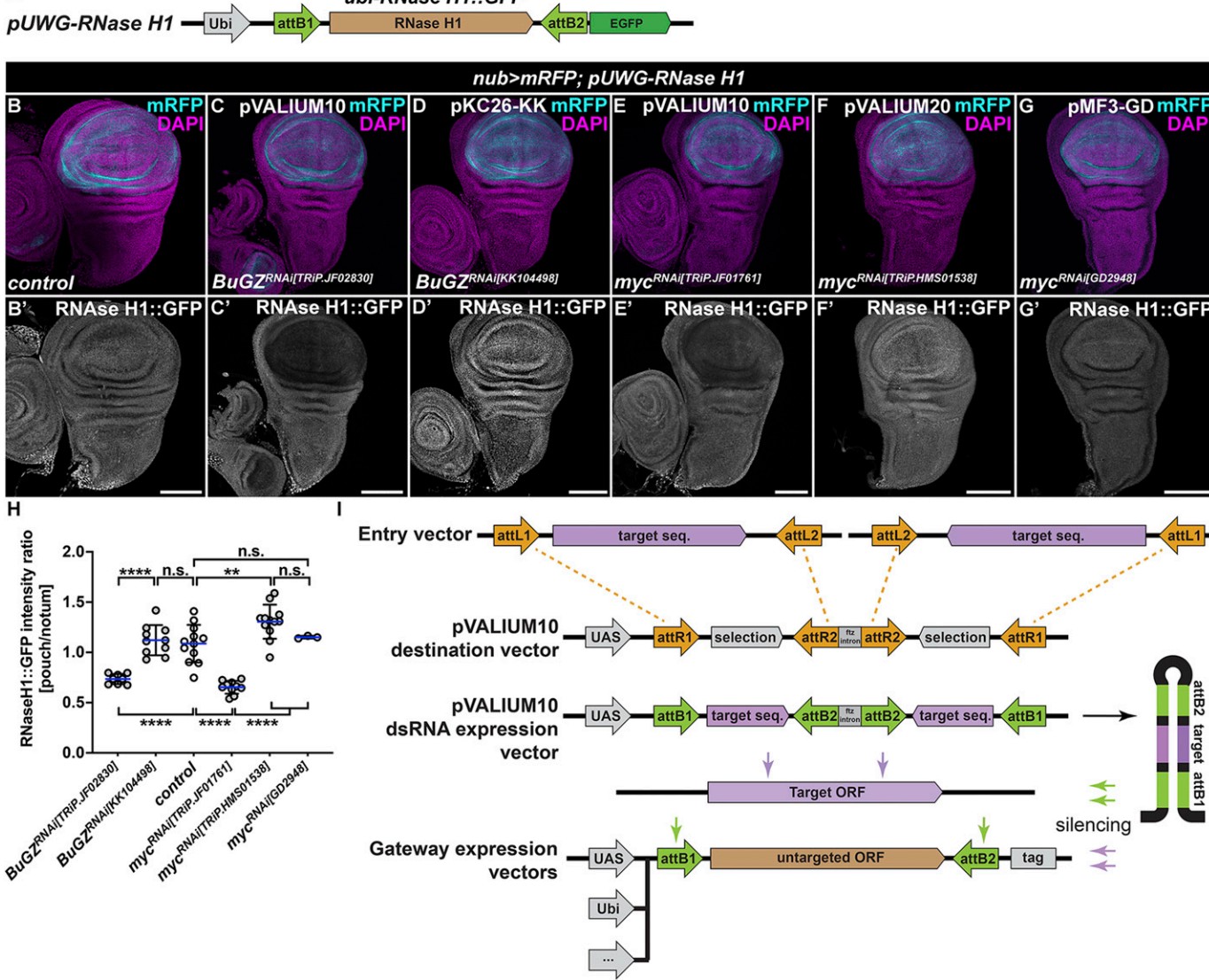

**Figure 2. pVALIUM10-based lines cause off-target silencing of the RNase H1::GFP transgenic reporter.**
**(A)** Schematic representation of the transgenic *pUWG-RNase H1* reporter construct expressing the C-terminally GFP-tagged RNase H1::GFP protein from the polyubiquitin (*ubi*) promoter. The *RNase H1* coding sequence is flanked by attB1 and attB2 sites generated via Gateway-mediated recombination catalyzed by the LR Clonase. **(B, C, D, E, F, G)** Immunostaining with a GFP-specific antibody revealed a marked reduction of the RNase H1::GFP levels in the nubbin domain (mRFP, light blue), where pVALIUM10-based *BuGZ^RNAi[TRiP.JF02830]* (C) and *myc^RNAi[TRiP.JF01761]* (E) RNAi lines were expressed using the *nubbin-Gal4, UAS-myr-mRFP* driver (*nub > mRFP*). No such down-regulation of the RNase H1::GFP signal was observed in control WD (B) and after expression of KK-based *BuGZ^RNAi[KK104498]* (D), pVALIUM20-based *myc^RNAi[TRiP.HMS01538]* (F), or GD-based *myc^RNAi[GD2948]* (G). Micrographs show projections of multiple confocal sections of WDs dissected from third instar larvae 7 d AEL. Nuclei were counterstained with DAPI (magenta). Scale bars: 100 μm. **(H)** Quantification of the RNase H1::GFP signal depicted as a relative intensity ratio between the pouch and the notum region. Statistical significance was determined by one-way ANOVA with Tukey's multiple comparison test; n ≥ 3; **$P < 0.01$, ****$P < 0.0001$, and n.s., non-significant. **(I)** Generation of pVALIUM10 RNAi lines relied on a LR Clonase–mediated recombination of dsRNA fragment from an entry vector to the pVALIUM10 destination vector (Ni et al, 2009). The dsRNA hairpin produced from the pVALIUM10 dsRNA expression vector is the source of siRNA species targeting not only the transcript of interest (target ORF) but also attB1 and attB2 sites present in any of the Gateway expression vectors generated via the same recombination mechanisms. Source data are available for this figure.

Unexpectedly, we observed a reduction in RNase H1::GFP signal when expressing *BuGZ^RNAi[TRiP.JF02830]* but not *BuGZ^RNAi[KK104498]* line (Fig 2C, D, and H). Strikingly, similar down-regulation of RNase H1::GFP was caused by pVALIUM10-based *myc^RNAi[TRiP.JF01761]* but not *myc^RNAi[TRiP.HMS01538]* or *myc^RNAi[GD2948]* derived from pVALIUM20 and pMF3 vectors, respectively (Figs 1D and 2E–H). All three *myc* RNAi lines have been shown effective by others (Aughey et al, 2016; Kim-

Yip & Nystul, 2018; Rust et al, 2018; Ledru et al, 2022). These results indicated that the tested pVALIUM10 TRiP RNAi lines but not pVALIUM20, KK, or GD lines may cause undesirable silencing of the RNase H1::GFP transgene. We further speculated that dsRNA expressed from pVALIUM10 contains segments unrelated to the target gene-specific sequences that trigger this unexpected off-target effect.

### pVALIUM10 RNAi and Gateway expression plasmids share attB recombination sites

To identify the potential target for dsRNA-mediated knockdown of the RNase H1::GFP transgene by pVALIUM10 TRiP RNAi, we compared the plasmid sequences. We found that attB1 and attB2 sites are shared between the Gateway-based expression vectors, including pUWG, and pVALIUM10 TRiP RNAi plasmids. attB1 and attB2 sites are 25 bp in length and generated by LR Clonase–mediated recombination between attL and attR sequences. Given the repetition of attB1 and attB2 sites, flanking the passenger and guide target sequence fragments in pVALIUM10 TRiP RNAis, these sequences are transcribed and could contribute to the dsRNA stem–loop substrate for Dicer. Based on these results, we hypothesized that dsRNAs generated from attB1 and attB2 could target, and thus silence, Gateway-based transgenes via attB sites (Fig 2I).

### pVALIUM10 RNAi lines silence Gateway transgenes via attB sites

To test our hypothesis, we cloned the *mCherry* coding sequence into the pUWG Gateway expression vector (pUWG-*mCherry*) allowing the ubiquitous expression of *mCherry* transcript flanked by the attB1 and attB2 sites. The pUWG-*mCherry* vector then served as a template for overlap PCR to generate the pUWG$^{ΔattB}$-*mCherry* construct that lacks attB sites (Fig 3A). Both plasmids were used to establish transgenic fly lines through a standard *Drosophila* germline transformation method (Rubin & Spradling, 1982). The susceptibility or resistance of *ubi-mCherry* (pUWG-*mCherry*) and *ubi$^{ΔattB}$-mCherry* (pUWG$^{ΔattB}$-*mCherry*) transcripts to RNAi-mediated silencing was then tested by driving various RNAi lines under the control of the *nub-Gal4* driver. Immunostaining of the third instar wing imaginal discs using an anti-RFP antibody revealed an identical pattern of ubi-mCherry and ubi$^{ΔattB}$-mCherry transgenes in the control (*nub>*) background (Fig 3B and C). In contrast, levels of ubi-mCherry but not ubi$^{ΔattB}$-mCherry transgenic protein were markedly reduced in the wing pouch region relative to the rest of the tissue after overexpression of *BuGZ* and *myc* pVALIUM10 TRiP RNAi lines (Fig 3D, E, H–J, and M), whereas no significant changes were observed when *BuGZ* and *myc* were knocked down using KK and pVALIUM20 RNAi lines, respectively (Fig 3F–H and K–M). Importantly, the attB site–dependent silencing of ubi-mCherry was recapitulated using an additional set of pVALIUM10 TRiP RNAi lines targeting *Atf3*, *Lip4*, *yki*, *DH44*, *ftz-f1*, *mago*, *brat*, and *Actβ* genes (Fig S1).

Finally, we tested whether pVALIUM10 TRiP RNAi lines may also interfere with UAS-inducible attB-containing transcripts, such as transgenic fly ORFs expressed from the Gateway destination vector pGW-HA.attB (Bischof et al, 2014). To this end, we assessed levels of the SmD3::3xHA protein in the presence of the pVALIUM10 or KK BuGZ-RNAi line. In contrast to an enrichment of SmD3::3xHA protein and positive punctae in the wing pouch cells of control (*nub>*) and *BuGZ$^{RNAi[KK104498]}$* expressing wing discs, the overall levels and the number of positive punctae were noticeably reduced in *nub > BuGZ$^{RNAi[TRiP.JF02830]}$* cells (Fig 3N–Q). These results suggest that the dsRNA hairpin produced from pVALIUM10 RNAi vectors generates attB1- and attB2-siRNAs, which guide destruction of transcripts containing such target sites. This off-target silencing via attB sites is specific to RNAi lines based on pVALIUM10. The absence of attB1 and

attB2 sites renders the Gateway-derived transcript resistant to unintended pVALIUM10 TRiP RNAi silencing.

## Discussion

The use of transgenic RNAi to achieve developmental stage- and tissue-specific knockdown of candidate genes and to perform genome-wide screening in *Drosophila* has become commonplace in the past two decades (Dietzl et al, 2007; Port et al, 2011; Blake et al, 2017; Hu et al, 2017). Both our knowledge of the RNAi mechanism and the tools to generate new types of RNAi constructs efficiently have considerably evolved in recent years. As a result, several independent transgenic RNAi stock collections were established, including Vienna *Drosophila* Stock Center (VDRC, 23,411 RNAi lines; https://stockcenter.vdrc.at/control/main), Bloomington *Drosophila* Stock Center (BDSC, 13,698 RNAi lines; https://bdsc.indiana.edu/), and the National Institute of Genetics (NIG, 12,365 RNAi lines; https://shigen.nig.ac.jp/fly/nigfly/) at the time of writing. Yet, in spite of being a broadly used technique, transgenic manipulation of gene expression can still have unforeseen consequences. A large portion of the RNAi lines from the VDRC KK collection were found to produce phenotypes because of a secondary insertion and consequent ectopic expression of the *tiptop* gene (Green et al, 2014; Vissers et al, 2016). More recently, van der Graaf and colleagues reported that transgenes inserted in the commonly used *attP40* landing site on the second chromosome can decrease the expression of Msp300, a member of the LINC complex, and hence lead to nuclear size, shape, and positioning phenotypes (van der Graaf et al, 2022 Preprint).

Here, we demonstrate that the pVALIUM10-based RNAi lines cause off-target silencing of ORFs expressed from Gateway expression vectors or potentially any transcripts containing attB1 and attB2 sites. We show that the removal of these sites abrogates the undesirable knockdown. Given the popularity of the Gateway expression vectors to generate transgenic protein reporters and tagged overexpression constructs, our results suggest that the simultaneous use of pVALIUM10 TRiP RNAi may lead to experimental artifacts unrelated to the investigated biological process.

Of note, in the original publication of pVALIUM10 vectors, the authors tested for potential off-target effects by comparing the hairpin sequence against the *Drosophila* genome (Ni et al, 2009), which does not contain endogenous 21-nt stretches equivalent to the attB1 and attB2 sequences. However, as the transgenic toolkit in *Drosophila* is ever expanding, each additional transgene has the possibility to produce a synthetic effect with the existing RNAi lines. Hence, we suggest that the researchers validate the type of the TRiP RNAi line or the expression vector sequence before commencing their experiments. As an example, we do not recommend combining RNAi based on pVALIUM10 with the 3xHA-tagged overexpression lines from FlyORF (https://flyorf.ch) (Bischof et al, 2014) or the fly-FUCCI (fluorescent ubiquitination-based cell cycle indicator) lines, which allow real-time visualization of the cell cycle dynamics by the expression of two fluorescently-tagged degrons (Zielke et al, 2014). In both cases, the transgenes were generated via Gateway reaction and contain attB sites flanking the inserted ORF (Bischof et al, 2014;

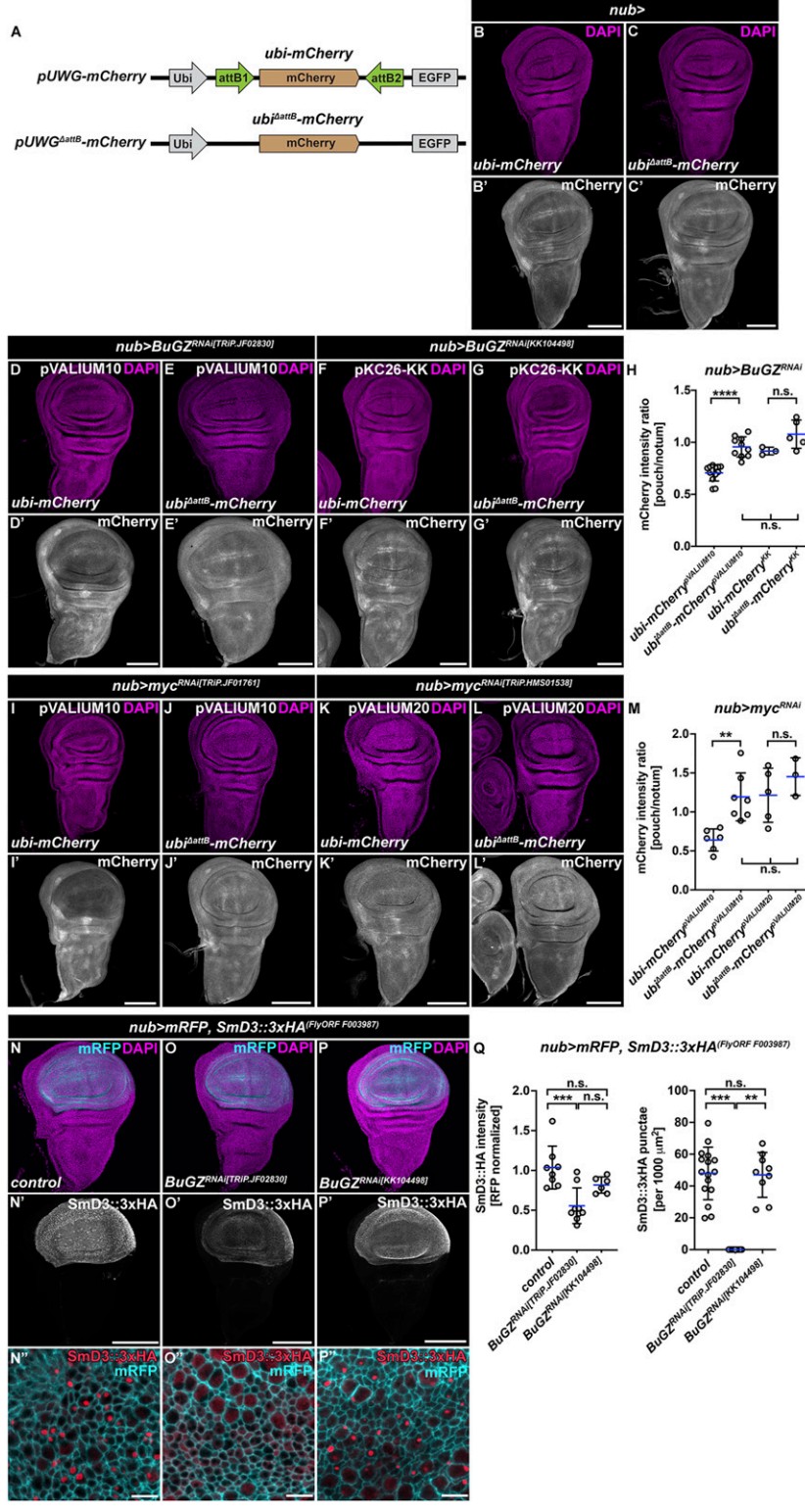

**Figure 3. pVALIUM10 RNAi lines silence Gateway transgenes via attB1 and attB2 sites.**
**(A)** Schematic of the transgenic *pUWG-mCherry* and mutant *pUWG*^*ΔattB*^*-mCherry* reporter constructs. The mCherry is expressed from the polyubiquitin (*ubi*) promoter. **(B, C)** *ubi-mCherry* and mutant *ubi*^*ΔattB*^*-mCherry* reporters showed a similar expression pattern in the wing imaginal discs. **(D, E, F, G, H, I, J, K, L, M)** *ubi-mCherry* was efficiently down-regulated by pVALIUM10 *BuGZ*^*RNAi[TRiP.JF02830]*^ and *myc*^*RNAi[TRiP.JF01761]*^ expressed under the control of the *nubbin-Gal4* driver (*nub>*) relative to the rest of the WD (D, H, I, M). **(E, H, J, M)** In contrast, the absence of attB1 and attB2 sequences rendered the *ubi*^*ΔattB*^*-mCherry* reporter resistant to pVALIUM10 RNAi–mediated silencing (E, H, J, M). **(F, G, H, K, L, M)** Neither KK-based (*BuGZ*^*RNAi[KK104498]*^) nor pVALIUM20-based (*myc*^*RNAi[TRiP.HMS01538]*^) RNAi lines affected mCherry levels expressed from the unmodified or mutant reporter constructs (F, G, H, K, L, M). Micrographs show projections of multiple confocal sections of WDs dissected from third instar larvae 7 d AEL that were immunostained with an anti-mRFP antibody (white). Nuclei were counterstained with DAPI (magenta). Scale bars: 100 $\mu$m. **(H, M)** Quantification of the mCherry signal depicted as a relative intensity ratio between the pouch and the notum region (H, M). Superscript text in each group denotes the type of RNAi used. Statistical significance was determined by one-way ANOVA with Tukey's multiple comparison test; n ≥ 3; \*\*$P < 0.01$, \*\*\*\*$P < 0.0001$, and n.s., non-significant. **(N, O, P, Q)** Expression of pVALIUM10 *BuGZ*^*RNAi[TRiP.JF02830]*^ (O) in the wing pouch using the *nubbin-Gal4, UAS-mRFP* driver (*nub > mRFP*) reduces SmD3::3xHA levels compared with control (N) and KK-based *BuGZ*^*RNAi[KK104498]*^ (P). Note foci of SmD3::3xHA protein in control (N") and *BuGZ*^*RNAi[KK104498]*^ (P") cells that are not present in *BuGZ*^*RNAi[TRiP.JF02830]*^-expressing cells (O"). Micrographs show projections of multiple or single (N", O", P") confocal sections of WDs dissected from third instar larvae 7 d AEL that were immunostained with an anti-HA antibody (white, N', O', P'; red, N", O", P"). Nuclei were counterstained with DAPI (magenta). Scale bars: 100 or 5 $\mu$m (N", O", P"). **(Q)** Quantification of the SmD3::3xHA signal intensity (Q, left) and SmD3::3xHA-positive punctae (Q, right) in the pouch region. Statistical significance was determined by one-way ANOVA with Tukey's multiple comparison test and Kruskal–Wallis test, respectively; n ≥ 6; \*\*$P < 0.01$, \*\*\*$P < 0.001$, and n.s., non-significant.
Source data are available for this figure.

*Zielke et al, 2014*), which can lead to off-target silencing of the transcript. Furthermore, this finding reinforces the idea that RNAi experiments should adhere to the standards established in the past decades, including investigating possible off-target effects to

eliminate knockdown of the unintended gene, using multiple independent RNAi lines to recapitulate phenotypes, and confirming the knockdown of the targeted gene products (Echeverri et al, 2006; Green et al, 2014; Kaya-Çopur & Schnorrer, 2016).

## Materials and Methods

### Fly stocks and husbandry

The following *Drosophila* strains were used: *w[1118]* (BDSC; RRID: BDSC_3605), *nubbin-Gal4, UAS-myr-mRFP* (*nub > mRFP*), *BuGZ[FlyFos[v318366]]* (VDRC), *BuGZ[RNAi[TRiP.JF02830]]* (BDSC; RRID:BDSC_27996), *BuGZ[RNAi[KK104498]]* (VDRC), *Lip4[RNAi[TRiP.HMO5136]]* (BDSC; RRID:BDSC_28925), *myc[RNAi[TRiP.JF01761]]* (BDSC; RRID:BDSC_25783), *myc[RNAi[TRiP.HMS01538]]* (BDSC; RRID:BDSC_36123), *myc[RNAi[GD2948]]* (VDRC), *Actβ[RNAi[TRiP.JF03276]]* (BDSC; RRID:BDSC_29597), *ftz-f1[RNAi[TRiP.JF02738]]* (BDSC; RRID:BDSC_27659), *yki[RNAi[TRiP.JF03119]]* (BDSC; RRID:BDSC_31965), *Atf3[RNAi[TRiP.JF02303]]* (BDSC; RRID:BDSC_26741), *DH44[RNAi[TRiP.JF01822]]* (BDSC; RRID:BDSC_25804), *brat[RNAi[TRiP.HM05078]]* (BDSC; RRID:BDSC_28590), *mago[RNAi[TRiP.HM05142]]* (BDSC; RRID:BDSC_28931), *SmD3::3xHA[FlyORF F003987]* (FlyORF), and *ubi-RNase H1::GFP*, *ubi-mCherry*, and *ubi[ΔattB]-mCherry* (this study). All *Drosophila* stocks are listed in Table S1. All crosses were set up and maintained at 25°C, unless specified otherwise, on a diet consisting of 0.8% agar, 8% cornmeal, 1% soymeal, 1.8% dry yeast, 8% malt extract, and 2.2% sugar–beet syrup, which was supplemented with 0.625% propionic acid and 0.15% Nipagin. *w[1118]* line was used as a control.

### Generation of plasmids and transgenic lines

The ORF of *rnh1* (FBgn0023171) was amplified from *Drosophila melanogaster* cDNA (*w[1118]*) using primers harboring NotI and SalI restriction sites. The stop codon was excluded to allow C-terminal tagging of the protein. The ORF was cloned into the Gateway pENTR 4 Dual Selection vector (Cat# A10465; Thermo Fisher Scientific). The ORF was then recombined into the pUWG vector (Cat# 1284; DGRC) using the LR Clonase II enzyme mix according to manufacturer's instructions (Cat# 11791020; Thermo Fisher Scientific). The coding sequence of *mCherry* (GenBank: AY678264.1) was amplified with specific primers, cloned into the pENTR 4 Dual Selection vector (Cat# A10465; Invitrogen) including a stop codon, and subsequently recombined into the pUWG vector (pUWG-*mCherry*). Overlap PCR strategy was employed to generate the pUWG[ΔattB]-*mCherry* construct lacking attB sites resulting from the LR recombination. Three separate PCRs were performed to produce fragments for assembly using the original pUWG-*mCherry* plasmid as a template. The forward primer of the first primer pair contained a XhoI restriction site, whereas the reverse primer contained a XbaI restriction site. Partial complementarity of the reverse and forward primers used to synthesize the downstream fragment enabled the exclusion of attB sites in the final product. The final PCR product was ligated into the pUWG-*mCherry* plasmid digested with XbaI and XhoI. Transgenic fly lines with random insertions of the constructs were generated by standard P-element–mediated transformation using the *w[1118]* *Drosophila* strain. All primers and plasmids are listed in Table S2.

### Tissue dissection and immunostaining

Wing imaginal disks dissected from third instar *Drosophila* larvae (7 d after egg laying) were fixed for 25 min with 4% paraformaldehyde in PBS containing 0.1% Triton X (PBS-T) at room temperature. Fixed tissues were washed three times with PBS-T. Primary antibodies were diluted in blocking buffer (PBS-T with 0.3% BSA), and tissues were stained overnight at 4°C. The following primary antibodies were used: rabbit anti-mRFP (1:500, Cat# PM005, RRID:AB_591279; MBL International), goat anti-GFP (1:500, Cat# ab6673, RRID:AB_305643; Abcam), and rabbit-anti HA (1:1,000, Cat# ab9110, RRID:AB_307019; Abcam). After washing, the samples were incubated with the corresponding Alexa Fluor 488 (1:2,000, Cat# A-11034, RRID:AB_2576217; Thermo Fisher Scientific)–, Cy2 (1:2,000, Cat# 705-225-147, RRID:AB_2307341; Jackson ImmunoResearch Labs)-, or Cy3 (1:2,000, Cat# 711-165-152, RRID:AB_2307443; Jackson ImmunoResearch Labs)-conjugated secondary antibodies overnight at 4°C, and washed and counterstained with DAPI (1:1,000 dilution of 5 mg/ml stock, Cat# 6335.1; Carl Roth GmbH) to visualize nuclei. Tissues were mounted on glass slides in DABCO–Mowiol 4-88 (Cat# D2522 and Cat# 81381; Sigma-Aldrich).

### Image acquisition and processing

Confocal images and stacks were acquired with an Olympus FV1000 confocal microscope equipped with 20× UPlan S-Apo (NA 0.85), 40× UPlan FL (NA 1.30), and 60× UPlanApo (NA 1.35) objectives. Maximum Z-projections were generated from consecutive sections taken at 1.4-µm steps using the FluoView 1000 software (Olympus) (RRID: SCR_014215) and Fiji (https://fiji.sc/) (RRID: SCR_002285). Final image processing, including panel assembly, and brightness and contrast adjustments were performed in Adobe Photoshop CC (Adobe Systems, Inc.) (RRID:SCR_014199). The schemes were generated in Adobe Illustrator CC (Adobe Systems, Inc.) (RRID:SCR_010279) and BioRender.com (RRID:SCR_018361; BioRender).

### Quantification of mCherry, RNase H1::GFP, and SmD3::3xHA signal

Fluorescent signal intensity was quantified with Fiji (https://fiji.sc/) (RRID:SCR_002285). For mCherry and RNase H1::GFP, a square selection was created in the middle of the pouch region, and the mean intensity was measured; then, the same selection was moved to the middle of the notum, and the measurement was repeated. The two values were divided to produce a pouch/notum intensity ratio for each wing disc. For SmD3::3xHA, the mean intensities of SmD3::3xHA and mRFP signals in the pouch region were measured and divided to produce a normalized value for each wing disk. Zoom images of the nubbin domain of independent WDs were used to count the number of SmD3::3xHA-positive punctae per 1,000-µm$^2$ area. Statistical significance was determined by one-way ANOVA with Tukey's multiple comparison test (signal intensity) or Kruskal–Wallis test (punctae) in GraphPad Prism (RRID: SCR_002798).

## Supplementary Information

# Acknowledgements

We thank Norbert Perrimon for discussions and comments on the article. We thank the Bloomington *Drosophila* Stock Center (BDSC, Bloomington, IN, USA), the Vienna *Drosophila* Resource Center (VDRC, Vienna, Austria), the *Drosophila* Genomic Resource Center (DGRC, supported by NIH grant 2P40OD010949, Bloomington, IN, USA), and the Zurich ORFeome Project (FlyORF, Zurich, Switzerland) for fly stocks and plasmids. We are grateful to Tina Bresser for generation of transgenic lines and Nils Teuscher for fly stock maintenance and technical assistance. This work was funded by the Deutsche Forschungsgemeinschaft (DFG, German Research Foundation) under Germany's Excellence Strategy – CECAD (EXC 2030 – 390661388).

## Author Contributions

D Stanković: resources, formal analysis, validation, investigation, visualization, methodology, and writing—original draft, review, and editing.
G Csordás: conceptualization, formal analysis, validation, methodology, and writing—original draft, review, and editing.
M Uhlirova: conceptualization, resources, data curation, formal analysis, supervision, funding acquisition, validation, investigation, visualization, methodology, and writing—original draft, review, and editing.

## Conflict of Interest Statement

The authors declare that they have no conflict of interest.

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
