## [Reviewer comments · Life Science Alliance]

Life Science Alliance

Drosophila pVALIUM10 TRiP RNAi lines cause undesired silencing of Gateway-based transgenes

Dimitrije Stankovic, Gabor Csordas, and Mirka Uhlirova

DOI: <https://doi.org/10.26508/lsa.202201801>

Corresponding author(s): Mirka Uhlirova, University of Cologne, CECAD Cologne

Review Timeline:

Submission Date:	2022-11-03
Editorial Decision:	2022-11-08
Revision Received:	2022-11-14
Accepted:	2022-11-16

Transaction Report:

Please note that the manuscript was reviewed at Review Commons and these reports were taken into account in the decision-making process at Life Science Alliance.

1. General Statements

We would like to thank the Reviewers for their valuable comments and constructive suggestions concerning our manuscript entitled "*Drosophila pVALIUM10 TRiP RNAi lines cause undesired silencing of Gateway-based transgenes*".

Please find below our responses to the Reviewers' questions and comments. We have revised the Manuscript following the Reviewers' suggestions. The changes in the Manuscript are indicated in blue.

Reviewer #1 (Evidence, reproducibility and clarity (Required)):

This manuscript by Uhlirova and colleagues identified an unwanted off-target effect in the pVALIUM10 TRiP RNAi lines that are commonly used in the fly community. The pVALIUM10 lines use long double-stranded hairpins and are useful vectors for somatic gene knock-down, hence they are widely used.

Here the authors find that any pVALIUM10 TRiP RNAi line can create the silencing of any transgenes that were cloned with the commonly used Gateway system. This is caused by targeting attB1 and attB2 sequences, which are also present in other *Drosophila* stocks including the transgenic flyORF collection. Hence, this is an important and useful information for the fly community that should be published quickly. All experiments are well documented and well controlled. I only have a few minor comments.

1. I recommend to mention the number of 1800 pVALIUM10 lines in Bloomington in the abstract rather than 11% to make clear that this is an important number of lines. (1800 of 13,698 lines in Bloomington are 13 and not 11 per cent?)

We now include the absolute number of pVALIUM10 lines in the manuscript abstract. The percentages have been corrected. Furthermore, we updated/corrected the total number of RNAi lines available from various stock centers in the Discussion, L153-L156.

The status on 23.10.2022

VDRC - 23,411 in total (12,934 GD lines; 9,674 KK lines; 803 shRNA lines)

Bloomington - 13,410 TRiP lines based on pVALIUM vectors (13,674 in total, including 264 non-pVALIUM, and 48 non-fly genes targeting lines)

NIG - 12,365 in total (5,676 TRiP lines; 7,923 NIG RNAi lines)

2. The authors may consider to call the 'unspecific' silencing effect an 'off-target' effect compared to intended 'on-target'. Such a nomenclature would be more consensus.

We changed the wording in the manuscript as suggested by the reviewer.

3. Ideally, all the imaging results in Figure 2 and 3 would be quantified. The simple 'V10' label in the Figure 3L and 3M is not the most intuitive, at least it took me a while to figure out what the authors compare.

The labeling in the charts has been changed. We now provide quantifications for the data shown in Figure 2 and 3.

4. Does the silencing also affect attR sequences? These are present after cassette exchange in many transgenes, most of the time not in the mRNA though, so it might not be so relevant.

A 22 nucleotide stretch of the attB2 site indeed shows a 100% match to the attL2 site. See the example alignment below. While we did not assess this possibility experimentally, attL sites would likely be susceptible to the same undesirable off-target silencing effects if present in the nascent or mature transcript.

```

attB1:1-25 1  ----- ACAAGTTTGTACAAAAAGCAGGCT ----- 25
attB2:1-25 1  ----- ACCCAGCTTTC TTGTACAAAGTGGT ----- 25
attL2:1-100 1  ----- ACCCAGCTTTC TTGTACAAAGTGGCATT A ----- 30
attL1:1-100 1  ----- CAAATAATGATTTTATTTT GACTGATAGTGACCTGTT CGT ----- 40
attR2:1-125 1  ----- CATAGTGACTGGATATGTTGTGTTTTACAGTATTATGTAGTCTGTTTTTATG ----- 53
attR1:1-125 1  ACAAGTTTGTACAAAAAGCTGAACGAGAAAACGTA AAAATGATATAAATATCAATATATTTAAATTAGATTTTGCA ----- 74

attB1:1-25 -----
attB2:1-25 -----
attL2:1-100 31 TAAGAAA GCATTGCTTATCAA TTTGTTGCAACGAACAGGTC ACTATCA GTCAAAA TAAAAATCATTATTTG ----- 100
attL1:1-100 41 TGCAACAAATTGATAAGCAATGCTT -TTTTATAATGCCA ACTTTGTACAAAAA -GCAGGCT ----- 100
attR2:1-125 54 CAAAA TCTAATTTAATATATTGATATTTATATCATTTTACGTTTCTCGTTCAGCTTTCTTGTACAAAGTGGT ----- 125
attR1:1-125 75 TAAAAACAGAC ----- TACATAATA -CTGTAAAAACACAACATATCCAG -TCAC -TATG ----- 125
    
```

Reviewer #1 (Significance (Required)):

This is an important and useful information for the fly community that should be published quickly.

Reviewer #2 (Evidence, reproducibility and clarity (Required)):

Stankovic, Csordas, and Uhlirova show that a specific subset of the TRiP RNAi lines available, namely the pVALIUM10 subset, can cause a knockdown of certain co-expressed transgenes that contain attB1 and attB2 sites. The authors demonstrate that while pVALIUM20 or Vienna KK lines for BuGZ or myc RNAi do not affect RNase H1:GFP expression, pVALIUM10 RNAi lines against BuGZ or myc significantly decrease expression of the RNaseH1:GFP transgene. The authors propose that, due to how these RNAi lines were constructed, the siRNA products could be targeting to attB1 and attB2 sites in transgenes that were made using similar methodology. To support this idea, they ubiquitously express mCherry transgenes encoding mRNAs either containing or lacking attB sites. They find that the knockdown of mCherry seen with several different pVALIUM10 RNAi lines is observed with the reporter mRNA containing attB sites, but is suppressed when the attB sites are removed from mCherry mRNA. They also find that the pVALIUM10 RNAi lines reduce the expression of the FlyORF transgene SmD3:HA.

The paper is very clearly written and the data presented is convincing.

Minor suggestions:

1. Figure 3 L+M The labels for the ubi-mcherry and ubi Δ attb-mcherry are switched in these graphs (i.e. ubi Δ attb-mcherry should be the one with a higher intensity in the pouch compared to the notum).
2. Figure 3M the labels don't match the RNAi lines used in H-K.

We corrected the labelling in the charts.

3. Figure 2 and 3. For the images of the transgenes, it seems as if the BuGZ RNAi line has a more drastic effect on RNaseH1 than mCherry, and vice versa for the myc RNAi lines. Did the authors notice a pattern with the decreased expression. Do some of the RNAi lines have a more consistent/severe impact, or might different transgenes be impacted to different extents?

Throughout the study and multiple experimental trials, we did not observe that the *BuGZ*^{RNAi} and *myc*^{RNAi} silencing efficiency would depend on whether the monitored reporter was RNase H1::GFP or mCherry. What has been reproducible is the differential impact of the three tested *myc*^{RNAi} lines on ubi-RNaseH1::GFP transgene. While pVALIUM10-based *myc*^{RNAi[TRIP.JF01761]} reduces RNaseH1::GFP signal Valium20 *myc*^{RNAi[TRIP.HMS01538]} enhances it and GD *myc*^{RNAi[GD2948]} has no effect, although the number of replicates for the latter is lower compared to the other tested lines. Why Valium20 *myc*^{RNAi[TRIP.HMS01538]} increases RNaseH1::GFP signal remains unclear for now. We would like to refrain from directly quantitatively comparing the effects of phenotypically different RNAi lines on differently tagged mRNAs/proteins. As the RNaseH1::GFP fusion protein is nuclear while the mCherry is cytoplasmic, their distinct subcellular localization and/or turnover rate may give a different overall impression on the change in fluorescence intensity (Boisvert *et al*, 2012; Mathieson *et al*, 2018). Another confounding factor is the described roles of *Drosophila Myc* in regulating transcription, translation, and cell growth (Gallant, 2007).

4. Line 150 unnecessary comma after Both
Line 131 knockdown should be knocked down
Line 133 should be "using an additional"
Figure legend 1 wing disc should be at least written out when the abbreviation (WD) is first used.

We thank the reviewer for pointing these out, the relevant corrections were performed.

Reviewer #2 (Significance (Required)):

Overall, this manuscript is an informative reminder that RNAi lines can have weaknesses that have not yet been considered, and we appreciate the authors work to inform the fly community about this specific issue. These insights are crucial for fly labs to consider when planning experiments that will use the pVALIUM10 RNAi lines in combination with other transgenesis modalities. The manuscript also provides a cautionary note for the usage of similar resources in other model organisms.

Reviewer #3 (Evidence, reproducibility and clarity (Required)):

Summary:

In their manuscript "Drosophila pVALIUM10 TRiP RNAi lines cause undesired silencing of Gateway-base transgenes", Stankovic *et al*. describe off-target silencing of transgenes expressed from Gateway systems when expressed in transgenic RNAi drosophila lines from the VALIUM10 collection. Using fluorescence microscopy and immunostaining, the authors show that this unintended silencing is specific to VALIUM20 lines and is not observed with VALIUM20, KK or GD lines that also allow gene-specific RNAi silencing. This pleiotropic silencing

effect was observed in 10 different VALIUM₂₀ lines and affected Gateway-based transgene expressed from an ubiquitous promoter (poly-ubiquitin, ubi) or from Gal₄/UAS systems. Finally, the authors identify the molecular basis of VALIUM₂₀ pleiotropic silencing on Gateway transgenes as being due to the presence of short sequences used for PhiC₃₁-based recombination in the Gateway and the VALIUM systems, and could lead to the production of siRNAs against PhiC₃₁ recombination sites in VALIUM₁₀ lines. Using Gateway transgenes lacking the recombination sites (attB₁ and attB₂), the authors could abrogate silencing of the transgene in VALIUM₁₀ lines, confirming the recombination as shared targets between the Gateway and the VALIUM systems.

Major comments:

- The study is well designed and the key conclusions are convincing.
- However, the authors provide only fluorescence microscopy data to show decreased transgene expression. To confirm pleiotropic RNAi effect on Gateway transgenes in VALIUM₁₀, the authors should assess silencing with another technique. For instance, expression levels of proteins from Gateway transgenes could be measured by Western blot (e.g.: by assessing protein levels of GFP or other tags present in the Gateway transgenes).

In the manuscript, we present microscopy data as this is the typical use case for fluorescent reporters. The strength of the microscopy, in contrast to Western Blot or RT-qPCR approach, is that it allows us to directly compare the impact of RNAi silencing on cells that express the dsRNA transgene (cell-autonomous) to surrounding neighbor cells. The fluorescent imaging of WDs where all cells express the reporter construct, but only a subset of cells trigger RNAi-mediated silencing, provides spatial resolution and means for normalization while minimizing artifacts that can arise during tissue processing for WB and RT-qPCR. We provide data on GFP and HA-tagged transgenes, respectively, and untagged mCherry expressed from Gateway vectors under ubiquitin or UAS regulatory sequences with the explicit reason to show that the silencing effect is independent of the type of the protein tag or the expression regulator sequence.

In addition, the claim on line 141, "These results strongly indicate that the dsRNA hairpin produced from pVALIUM₁₀ RNAi vectors generates attB₁- and attB₂-siRNAs", should be modified. The authors only present fluorescence microscopy data to show decreased transgene expression and do not actually provide data on siRNA expression in the pVALIUM₂₀ lines. Therefore, with the current data, the authors should only say that their results suggest that the dsRNA hairpin produced from pVALIUM₁₀ RNAi vectors generates attB₁- and attB₂-siRNAs.

In order to substantiate their claim about pleiotropic RNAi effects from VALIUM lines on Gateway transgenes due to the production of attB₁- and attB₂-siRNAs, the authors should perform an experiment to show attB₁- and attB₂-siRNAs production in VALIUM₁₀ lines and not in VALIUM₂₀, KK or GD lines. Deep-sequencing analysis of siRNA (i.e.: miRNA-seq) from tissue expressing the corresponding RNAi transgenes would be an excellent approach to assess siRNA production in multiple samples at once. Alternatively, the authors could search published miRNA-seq datasets from VALIUM₁₀ and other RNAi lines to assess the presence of attB₁- and attB₂-siRNAs only in VALIUM₁₀ lines. This would be free and require only a few days of data mining and analysis, if such datasets exist already.

Another cheaper and faster approach (if lacking easy access to sequencing platform or bioinformatics capability) would be to perform small RNA northern blots analysis from fly tissues expressing VALIUM₁₀ vs VALIUM₂₀ (or KK or GD lines) and should take only a few days to do as described in doi: 10.1038/nprot.2008.67.

If such experiments or analyses cannot be performed, then the authors can only conclude that their data suggest that the unintended silencing of Gateway transgenes in VALIUM₁₀ is likely due to the production attB₁- and attB₂-siRNAs production.

Full Revision

We thank the reviewer for the valuable suggestions on experimental approaches to identify the exact interfering RNAs produced by the VALIUM₁₀-based RNAi constructs, which can be useful for controlling the specificity of knockdown of transgenes in studies using the resources mentioned in this report.

We believe the fluorescence micrographs and quantifications demonstrate the off-target silencing effects of pVALIUM₁₀-based RNAi lines on transgenic reporters generated using the Gateway LR cloning approach. Furthermore, we provide genetic evidence that removing the attB₁ and attB₂ sites from the reporter construct, which is otherwise identical to the original transgene (same promoter, same position of insertion, same genetic background), is sufficient to abolish the off-target effect. We would argue that the functional genetic experiments we performed with the original and mutated reporters represent the strongest possible evidence to confirm that silencing is taking effect via the attB sites.

As we do not attempt to detect siRNA complementary to attB₁/attB₂ sites directly, we have changed the statements in question as per the recommendation of the reviewer.

- The current data and methods are adequately detailed and presented, and the statistical analysis adequate.

Minor comments:

- The current manuscript does not have specific experimental issues.
- Prior studies are referenced appropriately
- Overall the text and figures are clear and accurate except for the following issues with Figure 3 and its legends
On lines 396, 397, 399 and 403, the authors refer to "wild-type" ubi-mCherry. This transgene directs the ubiquitous expression of a heterologous reporter gene and thus can not as "wild type". It could instead be referred to as the "original" or "unmodified" transgene.

We removed "wild-type" from the text.

Fig.3 L: the x-axis labels are wrong. Decrease in the mCherry intensity ratio is observed with the ubi-mCherry construct and not in the ubi Δ attB-mCherry, where the attB sequences thought to be targeted by the pVALIUM₁₀ have been deleted.

We corrected the Figure panels according Reviewer 1 and 3's observation.

- More space should be added between the first row of images (B-G), the second (H-L) and also the third (M-P) to avoid confusion between the labeling of the figures. Finally, to help contextualize their findings and gauging the extent of the risk of using VALIUM₁₀ lines in RNAi screen where a Gateway transgene is involved, the authors could provide information on the overlap between the VALIUM₁₀ collection and VALIUM₂₀, GD and KK collections. Knowing how many genes are uniquely targeted by VALIUM₁₀, could be helpful.

Of the TRiP pVALIUM-based RNAi stocks currently available in BDSC, 686 genes are targeted exclusively by pVALIUM₁₀ RNAi lines. Considering KK, GD and shRNA transgenic lines from VDRC and NIG RNAi collection, 17 genes remain unique targets for pVALIUM₁₀ lines.

TRiP pVALIUM-based lines (BDSC)

Reviewer #3 (Significance (Required)):

- The manuscript "Drosophila pVALIUM10 TRiP RNAi lines cause undesired silencing of Gateway-base transgenes" by Stankovic et al. is a technical study that sheds light on potential limitations of using common RNAi drosophila lines, namely the VALIUM10 collection.

- The study provides information about very specific genetic screens conditions in Drosophila, that are likely to be rare. A rapid Pubmed search with the following terms: "drosophila TRiP screen" returns only 11 citations, while a similar search with "drosophila CRISPR screen" returns 99 citations. This suggests that in vivo RNAi screen in Drosophila using TRiP RNAi collections might not be as common or powerful as CRISPR-based screens.

- The reported findings might be of interest mostly to a small group of scientists working with Drosophila melanogaster that specifically rely on VALIUM10 lines to perform in vivo RNAi screen in combination with Gateway transgene expression. This very specific combination of parameters is rare, since other RNAi fly stock collections exist (e.g.: VALIUM20, 21, KK, GD...). Furthermore, the advent of CRISPR tools that allows tissue-specific gene knock-out has led to the rapid expansion of CRISPR fly stock collections (<https://doi.org/10.7554/eLife.53865>).

Regardless of the limited scope of the study, this kind information is still valuable, albeit to a very limited audience.

- My relevant fields of expertise for this study are : insect RNAi, RNAi of RNAi screens and drosophila genetics.

We would like to raise some points concerning the above comments.

While TRiP-screen may not be an often-used keyword combination, the use of the TRiP lines is, in fact, ubiquitous in the *Drosophila* community. The tissue-specific RNA interference is still commonly utilized as a rapid, first-generation screening method that can be performed in a tissue-specific manner, representing one of the key advantages of the *Drosophila* model. To illustrate, since the submission of our manuscript a new study published by Rylee and co-workers investigated *Drosophila* pseudopupul formation by screening 3971 TRiP RNAi lines (Rylee *et al*, 2022). In contrast, genetic screens relying on mutant alleles usually require at least one additional cross, effectively doubling the time of the experiment. In addition, tissue-specific or temporarily restricted knockdown is sometimes required in screens, as full-body loss of function is often lethal or has developmental phenotypes incompatible with assessing gene function later in life.

The use of tissue-specifically driven Cas9 with integrated gRNA-expressing vectors is indeed becoming more common. However, this technique, much like RNA interference, is not without flaws. First, this produces knockout instead of knockdown, which means it has to be induced early in order for the resulting mutation to take effect. Otherwise, the remaining mRNA/protein may prevent the development of a phenotype. Second, the Cas9 must be titrated as high Cas9 levels have adverse phenotypes (Huynh *et al*, 2018; Meltzer *et al*, 2019; Poe *et al*, 2019; Port *et al*, 2014). Third, in our personal experience, as well as literature reports (Mehravar *et al*, 2019; Port & Boutros, 2022), indicate that the resulting phenotype can produce mosaics in the tissue.

Although the combination of Gateway-based reporters with TRiP-RNAi lines may seem like a fringe case, there are popular reporters that could be screening targets. Potentially the most well-known is the live cell cycle indicator fly-FUCCI system (Zielke *et al*, 2014), which allows the analysis of the cell cycle in real-time thanks to the expression of two fluorescently tagged degrons. As FUCCI transgenes were constructed with Gateway recombination, they represent targets of the pVALIUM10 TRiP lines. We now include the fly-FUCCI system as an example in addition to 3xHA-tagged FlyORF collection in the Discussion.

REFERENCES

- Boisvert FM, Ahmad Y, Gierlinski M, Charriere F, Lamont D, Scott M, Barton G, Lamond AI (2012) A quantitative spatial proteomics analysis of proteome turnover in human cells. *Mol Cell Proteomics* 11: M111 011429
- Gallant P (2007) Control of transcription by Pontin and Reptin. *Trends Cell Biol* 17: 187-192
- Huynh N, Zeng J, Liu W, King-Jones K (2018) A Drosophila CRISPR/Cas9 Toolkit for Conditionally Manipulating Gene Expression in the Prothoracic Gland as a Test Case for Polytene Tissues. *G3 (Bethesda)* 8: 3593-3605
- Mathieson T, Franken H, Kosinski J, Kurzawa N, Zinn N, Sweetman G, Poeckel D, Ratnu VS, Schramm M, Becher I *et al* (2018) Systematic analysis of protein turnover in primary cells. *Nature Communications* 9: 689
- Mehravar M, Shirazi A, Nazari M, Banan M (2019) Mosaicism in CRISPR/Cas9-mediated genome editing. *Developmental Biology* 445: 156-162
- Meltzer H, Marom E, Alyagor I, Maysel O, Berkun V, Segal-Gilboa N, Unger T, Luginbuhl D, Schuldiner O (2019) Tissue-specific (ts)CRISPR as an efficient strategy for in vivo screening in Drosophila. *Nature Communications* 10: 2113
- Poe AR, Wang B, Sapor ML, Ji H, Li K, Onabajo T, Fazliyeva R, Gibbs M, Qiu Y, Hu Y *et al* (2019) Robust CRISPR/Cas9-Mediated Tissue-Specific Mutagenesis Reveals Gene Redundancy and Perdurance in Drosophila. *Genetics* 211: 459-472
- Port F, Boutros M (2022) Tissue-Specific CRISPR-Cas9 Screening in Drosophila. In: *Drosophila: Methods and Protocols*, Dahmann C. (ed.) pp. 157-176. Springer US: New York, NY
- Port F, Chen HM, Lee T, Bullock SL (2014) Optimized CRISPR/Cas tools for efficient germline and somatic genome engineering in Drosophila. *Proc Natl Acad Sci U S A* 111: E2967-2976
- Rylee J, Mahato S, Aldrich J, Bergh E, Sizemore B, Feder LE, Grega S, Helms K, Maar M, Britt SG *et al* (2022) A TRiP RNAi screen to identify molecules necessary for Drosophila photoreceptor differentiation. *G3 Genes|Genomes|Genetics*: jkac257
- Zielke N, Korzelius J, van Straaten M, Bender K, Schuhknecht GFP, Dutta D, Xiang J, Edgar BA (2014) Fly-FUCCI: A versatile tool for studying cell proliferation in complex tissues. *Cell Rep* 7: 588-598

November 8, 2022

RE: Life Science Alliance Manuscript #LSA-2022-01801

Dr. Mirka Uhlirova
University of Cologne, CECAD Cologne
Institute for Genetics
Joseph-Stelzmann Str. 26
Cologne 50931
Germany

Dear Dr. Uhlirova,

Thank you for submitting your revised manuscript entitled "Drosophila pVALIUM10 TRiP RNAi lines cause undesired silencing of Gateway-based transgenes". We would be happy to publish your paper in Life Science Alliance pending final revisions necessary to meet our formatting guidelines.

- please upload your main and supplementary figures as single files; please upload your tables as editable doc or excel files or make sure that they are in the doc file of your main manuscript text
- please add a Running title, Summary Blurb, and a category for your manuscript to our system
- please add the Twitter handle of your host institute/organization as well as your own or/and one of the authors in our system
- please add the author contributions and a conflict of interest statement to the main manuscript text
- please add a separate section for your figure legends (main figures, supplementary figures and table legends) to the main manuscript text

A. FINAL FILES:

B. MANUSCRIPT ORGANIZATION AND FORMATTING:

Sincerely,

November 16, 2022

RE: Life Science Alliance Manuscript #LSA-2022-01801R

Prof. Mirka Uhlirova
University of Cologne, CECAD Cologne
Institute for Genetics
Joseph-Stelzmann Str. 26
Cologne 50931
Germany

Dear Dr. Uhlirova,

Thank you for submitting your Research Article entitled "Drosophila pVALIUM10 TRiP RNAi lines cause undesired silencing of Gateway-based transgenes". It is a pleasure to let you know that your manuscript is now accepted for publication in Life Science Alliance. Congratulations on this interesting work.

DISTRIBUTION OF MATERIALS:

Again, congratulations on a very nice paper. I hope you found the review process to be constructive and are pleased with how the manuscript was handled editorially. We look forward to future exciting submissions from your lab.

Sincerely,
